# Level of satisfaction to clinical learning environment and its associated factors among nursing students of public universities in Central Ethiopia

Birhanu Tesfaye[1], Adamu Birhanu[2], Jiregna Darega[3], Samuel Abate Walda[1]*, Meseret Robi Tura[1]

1 Department of Nursing, College of Medicine and Health Sciences, Ambo University, Ethiopia,
2 Department of Psychiatry, College of Medicine and Health Sciences, Ambo University, Ethiopia,
3 Department of Public Health, College of Medicine and Health Sciences, Ambo University, Ethiopia

* abatesamuelo@gmail.com

## Abstract

### Background

Students' satisfaction with clinical learning environment is a vital to evaluate teaching-learning process. However, there is a scarcity of information regarding the nursing students' satisfaction with clinical learning environment and associated factors in Ethiopia. The purpose of this study was aimed to assess level of satisfaction to clinical learning environment and its associated factors among nursing students of public universities in Central Ethiopia, 2022.

### Méthodes

An institution based cross-sectional study was conducted among 245 undergraduate nursing students found in Universities of central Ethiopia from September 1–30, 2022. A simple random sampling technique was applied to recruit study participants. Data were collected via pretested self-administered questionnaire. The data were entered into Epi data version 3.1 and exported to SPSS version 25 for analysis. Descriptive statistics like frequency, percentage, median and IQR were computed. Binary and multivariable logistic regression analysis was done. For measuring the strength of the association between the outcome and independent variables, adjusted odds ratios (AOR) along with 95% confidence interval (CI) was used. Finally, statistical significance was declared at p-value <0.05.

### Results

In this study, nearly half of nursing students, 49% [95% CI (42.6–55.4)] were satisfied with clinical learning environment. Students who did met their clinical learning

**Data availability statement:** All relevant data are within the manuscript.

**Funding:** The author(s) received no specific funding for this work.

**Competing interests:** The authors declare that they have no competing interests for the publication of this work.

**Abbreviations:** AAU, Addis Ababa University; AU, Ambo University; AOR, Adjusted Odd Ratio; CI, Confidence Interval; CLE, Clinical Learning Environment; CLES+T, Clinical Learning Environment, Supervision and Nurse Teacher Scale; CS, Clinical Setting; CMHS, College of Medicine and Health Science; COR, Crude Odd Ratio; HERQA, Higher Education Relevance and Quality Agency; IQR, Inter Quartile Range; KM, Kilometer; NT's, Nurse Teachers; SPSS, Statistical Package in Social Science; UKMMC, University Kebangsaan Malaysia Medical Centre; WM, Ward Manager

outcome [AOR = 2.742 (95%CI: 1.407–5.343)], Students those who met 3 times per week with clinical preceptors [AOR: 2.829 (95% CI: 1.428–5.602)] and students who perceived inadequate supporting staffs [(AOR: 0.136 (95%CI: (0.073–0.254)] were associated factors.

## Conclusion

Meeting clinical learning outcome, students' perception of staff supports and having a frequent contact of supervisor were an association with satisfaction statistically.

---

## Introduction

### Background

Clinical learning environment (CLE) is a collaborative scheme within the clinical setting (CS) that influences skill learning outcomes and the area where the theoretical dimension of the curriculum is integrated with the practical skills [1]. It consists of five dimensions which are the ward's pedagogical atmosphere; the supervisory relationship; the ward's leadership; the premises of nursing on the ward and the clinical preceptor's role [2]. Higher Education Relevance and Quality Agency (HERQA) also define CLE as clinical training facilities/practice sites which are comprises of preceptors, different wards, necessary resources for giving the students adequate clinical experience, and patients within clinical setting [3]. In nursing education, a favorable CLE is enhancing student's independence, critical thinking, problem-solving, ethics, safety practices, a sense of responsibility and to advance their competencies [4]. As a nursing education is skilled based profession, CLE is a bond between academic learning and the clinical setting in which students integrate theory and practice [5]. This is eventually influence positively on nursing care delivery to patients and the opportunity to practice nursing skills with adequate supervision and give the chance to students to learn without distraction [6]. The conduciveness of CLE affected by supervisory method and feedback, characteristic of clinical setting, the degree of compatibility to the clinical learning objectives, learning opportunities, the relationship of students with preceptor and clinical supportive staff, nursing care in the ward and students' expectations [7]. Nursing student's satisfaction is considered as a crucial influencing factor when evaluating the effectiveness of the CLE and it plays a key role in determining the educational system [2]. As well as it enhances students' motivation to accept their career roles, apply nursing skills competently and carry out their responsibilities precisely and skillfully [2,8].

### Statement of the problem

Assuring the quality and relevance of higher education is recognized as a priority agenda both in the Education Sector Development Program IV and the Growth and Transformation Plan of Ethiopia. The Higher Education Proclamation mandated the Higher Education Relevance and Quality Agency (HERQA) to ensure that higher education institutions (HEIs) provide high quality and relevant education and to establish

a reliable internal quality assurance (IQA) system [3]. Good CLE is required in preparing the nursing students to providing quality care to their patients during their work time and increases students 'professional responsibility; whereas unconducive CLE makes students 'practice difficulty leading to poor achievement of competences [6]. Increased public expectation for quality and ethical health care is necessitating changes in what health professionals are taught and how they are taught. On the other hand, the increasing need to train more health workers coupled with rapid expansion in medical knowledge presents a serious challenge to the quality of education of health professionals including nursing. Despite these challenges, many HEIs training healthcare providers do not have well-functioning quality systems that have been cascaded to the department level [3].

Satisfaction is one of an indicator of a good CLE and in the nursing education; the quality of the CLE affects significantly student's satisfaction [2]. However, there could be indications that there is dissatisfaction among nursing students and this could be disturbing their clinical learning and gaining of clinical skills. One of these indications could be poor practical results coming from the nursing training institutions [6]. Globally, various researchers found that unconducive CLE can delay the achievement of learning objectives, delay gaining of skills and dissatisfaction among nursing students [2,9–11]. A global report on health education sees educational institutions are crucial to transform healthcare systems, however there is a challenge with clinical supervision, support of students in the clinical area and student involvement in clinical practice as well as nursing care provided by the newly qualified nurses is not up to the required standards and the life of the patient is at risk [6,12]. Also, the study revealed that negative experiences with CLE delay the achievement of learning outcomes, leading to a shortage of competent professional nurses at national level [13]. In addition, nurse students found CLE unharmonious with learning because of a lack of an easy approach to clinical staff and inability to get supervisors' feedback and make them dissatisfied [1]. Also, students dissatisfied as staff often complained about students in the presence of patients and their families; which makes students confused and embarrassed, even losing the trust of the patient for implementation of the next procedures these make students inactive in the patient care and absent from practice [14]. Nursing students were poorly satisfied with assigned ward without considering their learning objectives and the large numbers of students present in the CS at the same time makes it difficult to supervise on practice and unable to do necessary procedures in ward [7]. This result in absent from clinical practice and UN met their clinical objectives [15]. There were actions taken by minister of education in Ethiopia to enhance student's clinical learning outcome by increasing year of study from three to four year and assigned preceptors with at least Bachelor's degree with minimum of 2 years of service in the specific area [3]. Still there is poorly competent students, and dissatisfied with their CLE. Nurse students dissatisfied with CLE during clinical training gain; insufficient knowledge, poor skills performance and unable to achieve their clinical learning goal and faced with theory practice gap, this result in increased dropout rates and risk for patient at the time of their future career [8]. There are various variables affect nursing student satisfaction with CLE include:- student perception and motivation, attitude of staff and preceptor, characteristics of ward, peer support, inadequate equipment and clinical supervision, types of clinical center, patients and their family [5,16], Also, satisfaction of nurse students is affected by frequency of supervision, duration, learning opportunities [2,17]. In addition, students have complained of dissatisfaction related to limited participation in patient care, inadequate support from clinical staff and there is disharmony between theory and practice [15]. Previous studies clarified that for improving teaching-learning process; nursing student satisfaction plays a significant role [18].

## Significances of the study/ Benefits of these findings for the education and preparation of nurses

The finding of this study will be help nursing department, minister of education and nurse association to understand the nursing student's satisfaction with clinical learning environment and factors that affect nurse students' satisfaction and to design strategy to enhance factors positively influence on nurses 'student satisfaction as well as improve factors that contribute for dissatisfaction with clinical learning environment. In addition, the knowledge generated from this study will be enriched literatures available on the issue and aid for further a researcher who has an interest in conducting researches on this topic.

**Study question**

1. What is the student level of satisfaction with the clinical learning environment?

2. What are the factors that are associated with nursing students' satisfaction with the clinical learning environment?

## Objectives of the study

### General objective

The general objective of this study was to assess the level of satisfaction with the clinical learning environment and its associated factors among nursing students of public universities in Central Ethiopia in 2022.

### Specific objectives

To assess level of satisfaction with the clinical learning environment among nursing students of public universities in Central Ethiopia, 2022.

To identify factors that are associated with nursing students' satisfaction with the clinical learning environment at public universities in Central Ethiopia, 2022.

## Materials and methods

This study was conducted at public Universities Health Sciences College found in Central Ethiopia; namely, Ambo University, Addis Ababa University, Selale University, Debre Birhan University and Arsi University. Ambo University (AU) is one of the 2nd generation public universities in Ethiopia; established in 1947 as college and become university 2008. It located in Ambo town Oromia region and 114KM far from Addis Ababa to west, which has 4 campuses. In this study, Ambo university main campus and Ambo University Weliso campus which are giving nursing education were selected. Addis Ababa University (AAU) is one of the oldest and first-generation public universities in Ethiopia and founded in 1950 as University College of Addis Ababa. In 1961 it was restructured and renamed Haile Selassie I university, and in 1975 it adopted its present name. AAU has fourteen campuses. The study was conducted at Tikur Anbessa Health Science College. Selale University is one of study area and located in Fiche town, Oromia region 114 km North from Addis Ababa and established 2016.It is one of the fourth generation public universities and has three campuses. The study was conducted at Abebech Gobena Campus. The other study area is Debre Birhan University which was established in 2007 and one of the 2nd generation public universities in Ethiopia. It is located in Debre Birhan town, Amhara region which is 130 km away from Addis Ababa to North East. It has two campuses and the study was conducted at Asrat Woldeyes health science campus. Another study area is Arsi University and one of the 3rd generation public universities located in the Arsi zone of Oromia region 126 km south from Addis Ababa and established 2014. It has three campuses and this study was done at health science college Asella campus. The study was conducted from September 1 to September 30, 2022.

**Study design: –** An institutional based cross-sectional study involving quantitative study was conducted

### Source population

All second year and third year undergraduate Nursing students who are studying at universities found in Central Ethiopia.

### Study population

All randomly selected second year and third year undergraduate nursing students who are studying at universities found in Central Ethiopia.

## Sampling unit

Individual Nursing student who was selected during the data collection period

## Inclusion

All undergraduate nursing students of second year and third year as well as those participated at least once in clinical practice.

Exclusion criteria
Fourth year nursing student due to educational leave at the time of the study.

## Sample size determination

The sample size was calculated using a single population proportion formula. Assuming 95%, confidence level and 5% margin of error and 50% was taken for satisfaction as proportion of nursing students satisfied with CLE. Where P = Population proportion (assumed at 0.50) Z = Level of confidence interval 95% = 1.96 d 2 = margin of error 5% with 95% confidence level n = $(1.96)2 * 0.5(1–0.5)/ (0.05)2$ = 384 using correction formula since less than 10,000. Nr = n/1+ (n/N) where n = desired sample size nr = calculated sample size N = Total population nr = 384/1+ (384/492) = 226 by considering a 10%, non-response rate and the final sample size is 249.

## Sampling technique and procedure

Stratified random sampling technique was used to stratify students according to universities and year of study. Then, the study participants from each university according to years of study were selected by simple random. According to the data obtained from department of selected universities; there are a total of 492 nurse students. Sample size was proportionately allocated for each university according to the number of nurse students per year of study. The calculated sample size was proportional allocation for each selected University. Then, simple random sampling method by using Lottery method was used to select the students with list of student's identification number got from registrar.

nc = n x Nc N Where, nc = the sample size of the $c^{th}$ university n = Total final sample size Nc = Is total population of the university N = Total population of study area, Ambo university main campus (77 × 249)/492 = 39 Ambo university weliso campus (70 × 249)/492 = 35 Addis Ababa University (78 × 249)/492 = 40 Selale university (59 × 249)/492 = 30 DebreBirhan university (131 × 249)/492 = 66 Arsi university (77 × 249)/492 = 39 Then sample size for each year of study per university was calculated.

ny = n x Neu N where, ny = sample size per year of study n = number of students per year of study Neu = sample size of each university N = Total population of each university Ambo university main campus 2nd year 52*39/77 = 26, 3rd year 25*39/77 = 13 Ambo university weliso campus 2nd year 49*35/70 = 25, 3rd year 21*35/77 = 10 Addis Ababa University 2 nd year 40*40/78 = 21, 3rd year 38*40/78 = 19 Selale university 2nd year 34*30/59 = 17, 3 rd year 25*30/59 = 13 Debre Birhan university 2nd year 49*66/131 = 25, 3 rd year 82*66/131 = 41, Arsi university 2nd year 50*39/77 = 25, 3rd year 27*39/77 = 14 Hence for the 249-sample size, Ambo university main campus contribute with 39, Ambo university weliso campus is 35, Addis Ababa University is 40, Selale university is 30, Debre Birhan university is 66 and Arsi university 39.

## Data collection procedure

Stratified random sampling technique was used to stratify students according to universities and year of study. Then, the study participants from each university according to years of study were selected by simple random. According to the data obtained from department of selected universities; there are a total of 492 Nurse Students. Sample size was proportionately allocated for each university according to the number of nurse students per year of study. Proportional allocation was conducted using the following formula. In this regard, simple random sampling method

by using Lottery method was used to select the students with list of student's identification number. nc = n x Nc N Where, nc = the sample size of the $c^{th}$ university n = Total final sample size Nc = Is total population of the $c^{th}$ university N = Total population of study area, Ambo university main campus (77 × 249)/492 = 39 Ambo university weliso campus (70 × 249)/492 = 35 Addis Ababa University (78 × 249)/492 = 40 Selale university (59 × 249)/492 = 30 DebreBirhan university (131 × 249)/492 = 66 Arsi university (77 × 249)/492 = 39 Then sample size for each year of study per university was calculated. 16 ny = n x Neu N where, ny = sample size per year of study n = number of students per year of study Neu = sample size of each university N = Total population of each university Ambo university main campus 2 nd year 52*39/77 = 26, 3rd year 25*39/77 = 13 Ambo university weliso campus 2nd year 49*35/70 = 25, 3rd year 21*35/77 = 10 Addis Ababa University 2nd year 40*40/78 = 21, 3rd year 38*40/78 = 19 Selale university 2nd ear 34*30/59 = 17, 3rd year 25*30/59 = 13 Debre Birhan university 2nd year 49*66/131 = 25, 3rd year 82*66/131 = 41, Arsi university 2nd year 50*39/77 = 25, 3rd year 27*39/77 = 14 Hence for the 249-sample size, Ambo university main campus contribute with 39, Ambo university weliso campus is 35, Addis Ababa University is 40, Selale university is 30, Debre Birhan university is 66 and Arsi university 39.

## Data collection methods

**Data collection tool.** A structured self-administered adopted clinical learning environment, supervision and Nurse Teachers scale (CLES + T) tool was used to collect data. This tool was developed and validated 2008 by Saarikoski [19] as well as validated recently in 2020 [20]. The questionnaire used to collect data consists of two parts. Part one is consist of 15 items which cover socio demographic factors, students' factors, students-staff factors and institution factors which developed by investigator from literatures. Part two is called Clinical learning environment, Supervision and Nurse teachers scale (CLES + T scale) which is adopted a self-reported questionnaire with 34 items cover clinical learning environment dimensions and it is classified into five sub-dimensions: supervisory relationship(8items); pedagogical atmosphere on the ward(9 items); role of the nurse teacher(clinical preceptor) in clinical practice(9items), leadership of the ward manager(4 items); premises of nursing on the ward(4 items) and it has possibility of the 5-step continuum scale use in all phases of the study are: (1) very dissatisfied, (2) dissatisfied (3) not sure, (4) satisfied, and (5) very satisfied [20]. There are three BSC for data collection and three MSC nurses for supervision.

## Study variables

**Dependent variable.** Level of satisfaction to clinical learning environment.
**Independent variable. Socio-demographic factors** (Sex, age, Year of study).
**Student factors** (met clinical objectives of the placement, late to clinical work and use mobile phone for social media). **Staff-student factors** (Frequency of meeting with preceptors, oriented to objective, students to perform procedure under supervision). **Institutional factors** (Last types of health institution clinical placement, Last ward clinical placement, adequate materials, adequate supporting staff, and payment during practice).

## Data quality control

The researcher has adopted the CLES+T tool, a pretest was conducted one week before actual data collection on 5% [12] of Nurse students from, Menelik health science college of Kotobe Metropolitan university, found in Addis Ababa before start the main study to evaluate the clarity, reliability and to estimate the time needed to fill tool. Also, in this study content validity is covered by assuring that item in the research questions covered the research objectives and the concepts in the conceptual framework. There are three data collector and three supervisors. One day's orientation was given to them on the study tool and the data collection approach. Furthermore, the overall data collection process was monitored by the supervisors and an investigator accordingly. From the previous study, it shows the instrument to measure nursing students' satisfaction with CLE of were valid and the overall reliable value with total Cronbach's alpha of the CLES+T in

Turkish was 0.94 [20] and in Ethiopia Wolkite university was 0.92 [5]. From the current study the tool was reliable for each sub-dimension of CLE which are Supervisory relationship was 0.855, 20 Pedagogical atmosphere on the ward was 0.777, Role of nurse teacher was 0.705, Leadership style of ward manager was 0.849 and Premises of nursing on the ward was 0.647 and the overall reliability of CLES +T tool was with a total Cronbach's alpha at 0.875.

## Methods of data processing and analysis

The collected data was checked its completeness, entered into using Epi Data version 3.1 software and exported to SPSS version 25 for analysis. Data were cleaned by running the frequency and cross-checking any missing variable from the hard copy. Descriptive statistics like frequency, percentage, median and IQR were computed. Binary logistic regression was used to check for the association between the outcome variable and the independent variables. Variables that yield p-value of <0.25 in bivariable analysis were considered as candidate for multivariable logistic regression analysis. For measuring the strength of the association between the outcome and independent variables, adjusted odds ratios (AOR) along with 95% confidence interval (CI) was used. The Hosmer-Lemeshow goodness of fit was considered to check model fitness. Accordingly, the result for Hosmer Lemeshow goodness of the model was adequately fit at p value of (0.406). Finally, statistical significance was declared at p-value <0.05 and the result of this study was summarized and displayed in text, tables, figures and graphs.

## 4.10. Ethical consideration/ethical statement

Ethical clearance was obtained from Ethical Review Board of Ambo University college of Medicine and Health Sciences Research Committee (RREC ref. no: PGC/415/2014) legal permission paper to carry out the study was secured with the copy of proposal. Written informed consent was obtained from each of the students who agreed to participate in the research after explaining the aim and the importance of the study. They were informed that they have the right to participate or not in the research. They were also assured about confidentiality of the obtained data and that it was used for the research purpose only. No name written on the paper, but the code had to be put in order to keep their confidentiality.

## Results

Socio-demographic characteristics of the study participants (n = 245). A total of 249 questionnaires were distributed and 245 students were completed and return the questionnaire which makes a response rate of 98.4%. The majority 136 (55.5%) were males and more than half of the participants 174 (71%) were in the age group of 21–24 years old with median (IQR) of 21(±2). Most of the respondents 139 (56.7%) were 2nd year students (Table 1).

**Table 1. Socio-demographic characteristics of study participants in Central Ethiopia, 2022.**

| Variables | Category | Frequency(n=245) | Percentage (%) |
|---|---|---|---|
| Sex of study participants | Male | 136 | 55.5 |
| | Female | 109 | 44.5 |
| Age of study participants in years | <20 | 64 | 26.1 |
| | 21-24 | 174 | 71.0 |
| | >25 | 7 | 2.9 |
| | Median(±QR) | 2(±2) | |
| Year of study | 2nd year | 139 | 56.7 |
| | 3rd year | 106 | 43.3 |

## Students' factors

Regarding student factors, the majority of participants 141(57.6%) were met their clinical learning outcome. Above half 168(68.6%) of participants were not late during their clinical attachment and 154(62.9%) were no use mobile phone for social media while their working time (Fig 1).

## Staff-Students factors

Concerning staff-students, the majority of participants 160(65.3%) were met 1–2 times per week with their clinical preceptor during their last clinical attachment which was followed by 69(28.2%) participants were met 3 times per week with their clinical preceptor. Less than half 108(44.1%) of study participants were performing procedures under their clinical preceptor but above half 158(64.5%) of study participants were oriented to their clinical objectives by their clinical preceptor (Table 2).

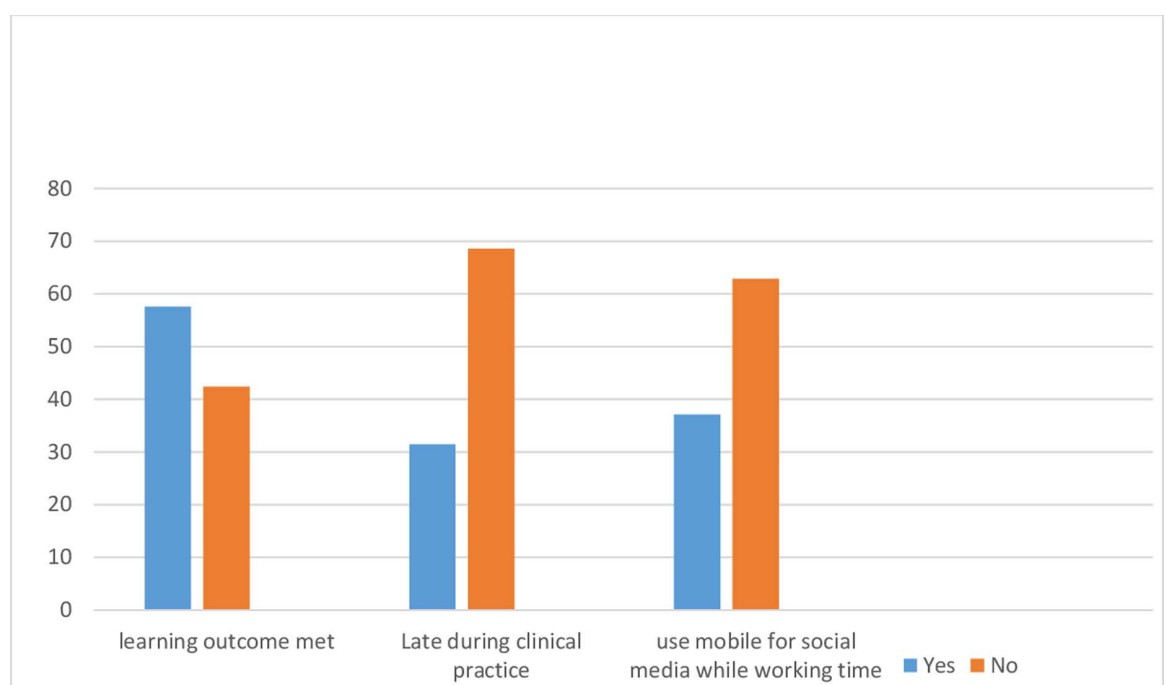

**Fig 1. Students' factors related to level of satisfaction to clinical learning environment among nursing students in Central Ethiopia, 2022.**

**Table 2. Staff-Students factors related to level of satisfaction to clinical learning environment in Central Ethiopia, 2022.**

| Variable | Category | Frequency(N) | Percentage (%) |
|---|---|---|---|
| Frequency met with clinical preceptor | 1 to 2 times per weak | 160 | 65.3 |
| | 3 times per weak | 69 | 28.2 |
| | Always | 16 | 6.5 |
| Performing procedure under clinical preceptor | Yes | 108 | 44.1 |
| | no | 137 | 55.9 |
| Did your preceptor orient to your objectives | Yes | 158 | 64.5 |
| | no | 87 | 35.5 |

## Institution factors

As institution factors, most of participants 196(80%) attached the clinical practice at the referral hospital in their last clinical placement site and 83(33.9%) participants attended a 26 surgical ward for their last clinical placement which was followed by medical ward 80(32.7%). Also, above half 137(55.9%) of the participants were reported that inadequate supporting staff in the clinical setting (Table 3).

## Level of nursing students' satisfaction to clinical learning environment

Possible scores on the CLES +T 34 items questions could range from 34 to 170 while in present study, the scores ranged from 50 to 163. The median for total nursing students' satisfaction with CLE after summing up all items was 108 (IQR=39). The overall score of the respondents who are satisfied with CLE was 49% [95% CI: 42.6, 55.4] (Fig 2).

## Students' level of satisfaction with sub-dimension of CLE

Among sub-dimensions of the clinical learning environment nursing students highly satisfied with Supervisory relationship (49.8%) and the Leadership of the ward manager was the lowest satisfied by nursing students (43.7%) (Fig 3).

## Factors associated with level of nursing students' satisfaction

In bivariate logistic regression analysis; meeting clinical objective of the placement, frequency of students meeting with their clinical preceptors, performing procedure under supervision of clinical preceptor, oriented of students about objective of placement by clinical preceptor, adequate number of staff assigned to support student during clinical attachment and payment for student during clinical attachment were significant associated factors and candidate for multivariable logistic regression analysis (Table 4).

**Table 3. Institution factors related to level of satisfaction to clinical learning environment in Central Ethiopia, 2022.**

| Variables | Category | Frequency(n=245) | Percentage |
|---|---|---|---|
| Types of health institution in which last clinical practice was held | Primary hospital | 27 | 11% |
| | General Hospital | 22 | 9% |
| | Referral Hospital | 196 | 80% |
| Types of wards of your last clinical placement | Medical Ward | 80 | 32.7% |
| | Surgical Ward | 83 | 33.9% |
| | Obstetrics and gynecological Ward | 24 | 9.8% |
| | Adult ICU | 15 | 6.1% |
| | Emergency ward | 19 | 7.8 |
| | Pediatrics Ward | 12 | 4.9 |
| | Others | 12 | 4.9 |
| Enough items in clinical setting | Yes | 103 | 42 |
| | No | 142 | 58 |
| Adequate supporting staff in clinical setting | Yes | 108 | 44.1 |
| | No | 137 | 55.9 |
| Payment during clinical practice | Yes | 70 | 28.6 |
| | No | 175 | 71.4 |
| How many do you paid per day | 50 | 19 | 7.8 |
| | 60 | 175 | 71.4 |
| | Above 100 | 51 | 20.8 |

## over all satisfaction of nursing students

☐ not satisfied
☐ satisfied

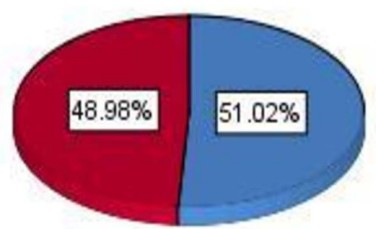

48.98%  51.02%

**Fig 2. Level of Satisfaction to clinical learning environment and its associated factors.**

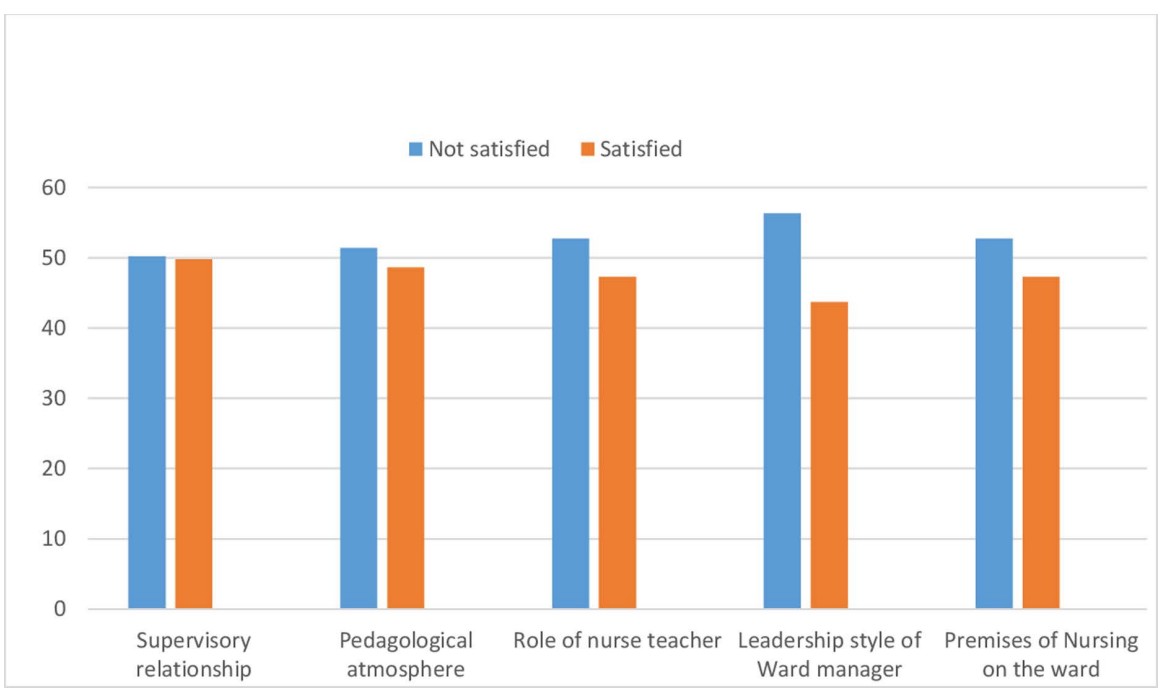

**Fig 3. Nursing students' level of satisfaction with sub dimension of CLE in Central Ethiopia.**

In multivariable logistic regression analysis, meeting clinical objective of the placement, frequency of students meeting with their clinical preceptors and adequate number of staff assigned to support student during clinical attachment were significantly associated with nursing student satisfaction with clinical learning environment at 95%CI with p-value of 0.05. Accordingly, Nursing students who met their clinical objectives of placement were about 2.742 times more likely to be satisfied with CLE as compared to those who not met [AOR = 2.742, 95%CI (1.407–5.343)]. Participants those who met 3

**Table 4. Bivariate logistic regression analyses of level of satisfaction among students in Central Ethiopia, 2022.**

| Variable | Category | Satisfied(%) | Not satisfied(%) | COR(95%CI) | p-value |
|---|---|---|---|---|---|
| Sex | Male | 73(53.7) | 63(46.3%) | 1 | |
| | Female | 47(43.1%) | 62(56.9%) | 0.654(0.394,1.086) | 0.101 |
| Age Category | <20 | 29(45.3%) | 35(54.7%) | 1 | 0.425 |
| | 21-24 | 89(51.1%) | 85(48.9%) | 1.264(0.711,2.264) | 0.404 |
| | >25 | 2(28.6%) | 5(71.4%) | 0.483(0.087,2.675) | |
| Year of study | 2nd | 69(49.6%) | 70(50.4%) | 1 | |
| | 3rd | 51(48.1%) | 55(51.9%) | 0.941(0.567,1.560) | 0.813 |
| Clinical learning outcome met | Yes | 88(62.4%) | 53(37.72%) | 3.736(2.181,6.398) | **0.000** |
| | No | 32(30.8%) | 72(69.2%) | 1 | |
| Late during your clinica practice | Yes | 39(50.6%) | 38(49.4%) | 1 | |
| | No | 81(48.2%) | 87(51.8%) | 0.907(0.529,1.556) | 0.723 |
| Use your mobile phone for social media while working time | Yes | 46(50.5%) | 45(49.5%) | 1 | |
| | No | 74(48.1%) | 80(51.9%) | 0.905(0.539,1.520) | 0.706 |
| How many time did you meet with your preceptors | 1 to 2 times per week | 68(42.5%) | 92(57.5%) | 1 | |
| During the last clinical placement | 3 times per week | 42(60.9%) | 27(39.1%) | 2.105(1.183,3,745) | **0.011** |
| | Always | 10(62.5%) | 6(37.5%) | 2.255(0.782,6.506) | **0.005** |
| | | 64(59.3%) | 44(40.7%) | 1 | |
| Do you your preceptors were following while you performing procedures | No | 56(40.9%) | 81(59.1%) | 0.475(0.285,0.794) | **0.005** |
| Did your preceptors orient you to your objectives | Yes | 85(53.8%) | 73(46.2%) | 1 | |
| | No | 35(40.2%) | 52(59.8%) | 0.578(0.340,0.983) | **0.043** |
| Types of health institution in which last clinical learning was held | Primary Hospital | 17(63%) | 10(37%) | 1 | |
| | Genera Hospital | 16(72.7%) | 6(27.3%) | 1.569(0.463,5.318) | 0.470 |
| | Refferal hospital | 87(44.4%) | 109(56.6%) | 0.47(0.205,1.077) | 0.074 |
| Types of wards of your last clinical placement | Medical Ward | 42(52.5%) | 38(47.5%) | 1 | |
| | Surgical Ward | 41(49.4%) | 42(50.6%) | 0.883(0.478,1.633) | 0.692 |
| | Obstetric & gyne ward | 9(37.5%) | 15(62.5%) | 0.543(0.213,1.384) | 0.201 |
| | Adult ICU | 7(46.7%) | 8(53.3%) | 0.792(0.262,2.391) | 0.679 |
| | Emergency ward | 9(47.4%) | 10(52.6%) | 0.814(0.299,2.217) | 0.688 |
| | Pediatrics Ward | 6(50%) | 6(50%) | 0.905(0.269,3.045) | 0.872 |
| | Others | 6(50%) | 6(50%) | 0.905(0.269,3.045) | 0.872 |
| There were enough items in clinical setting | Yes | 51(49.5%) | 52(50.5%) | 1 | |
| | No | 69(48.6%) | 73(51.4%) | 0.964(0.580,1.601 | 0.887 |
| There were adequate supporting staff in the clinical setting | Yes | 81(75%) | 27(25%) | 1 | |
| | No | 39(28.5%) | 98(71.5%) | 0.133(0.075,0.235) | 0.887 |
| Do you pay per diem during your clinical practice | Yes | 39(28.5%) | 34(48.6%) | 1 | |
| | No | 84(48%) | 91(52%) | 0.872(0.501,1.518) | **0.000** |
| Do you pay per diem during your clinical practice | Yes | 36(51.4%) | | 1 | |
| | No | 84(48%) | | 2.457(0.893,6.759) | 0.628 |
| If yes for Q14 how many do you paid per day | 50 | 6(31.6%) | 13(68.4%) | 82(46.9%) | 1 |
| | 60 | 93(53.1%) | 82(46.9%) | 2.457(0.893,6.759) | 0.082 |
| | Above 100 | 21(41.2%) | 30(58.8%) | 1.517(0.497,4.632) | 0.465 |

Table 5. Multivariable logistic regression analyses of factors associated with level of nursing student's satisfaction in Central Ethiopia, 2022.

| Variable | Category | satisfied | Not satisfied | COR | P-value |
|---|---|---|---|---|---|
| Clinical objective | Yes | 88(62.4%) | 53(37.72%) | 2.742(1.407,5.343) | 0.003 |
| | No | 32(30.8%) | 72(69.2%) | 1 | |
| frequency of latest met with clinical preceptors | 1 to 2times per week | 68(42.5%) | 92(57.5%) | 1 | |
| | 3times per week | 42(60.9%) | 27(39.1%) | 2.829(1.428,5.602) | 0.003 |
| Following while performing procedures | Yes | 10(62.5%) | 6(37.5%) | 1.744(0.447,6.807) | 0.423 |
| | No | 64(59.3%) | 44(40.7%) | 1 | |
| Oriented to objective | Yes | 56(40.9%) | 81(59.1%) | 0.827(0.406,1.684) | 0.600 |
| | No | 35(40.2%) | 52(59.8%) | 0.858(0.415,1.775) | 0.680 |
| Adequate supporting staff | Yes | 81(75%) | 27(25%) | 1 | |
| | No | 39(28.5%) | 98(71.5%) | 0.136(0.073,0.254) | 0.000 |
| Amount of paid during attachment | 50 | 6(31.6%) | 13(68.4%) | 1 | |
| | 60 | 93(53.1%) | 82(46.9%) | 2.622(0.766,8.974) | 0.125 |
| | Above 100 | 21(41.2%) | 30(58.8%) | 1.847(0.475,7.181) | 0.376 |

times per week with their clinical preceptors during latest clinical placement were 2.829 times more likely to be satisfied as compared with those who met 1–2 times per week [AOR = 2.829, 95%CI (1.428–5.602)]. Those participants who have inadequate supporting staff in the clinical setting were about 86.4% less likely to be satisfied than those whose clinical setting has adequate supporting staff [AOR = 0.136, 95%CI (0.073–0.254)] (Table 5).

## Discussion

This study aimed to assess the level of nursing students' satisfaction with clinical learning environment and associated factors at public universities, in Central Ethiopia. In the present study, 49% [95% CI (42.6–55.4] of nursing students were satisfied with the CLE. The finding in this study is similar to that reported by a study conducted in Egypt at Port Said University in which 52.9% [21] and in Nepal at Chitwan medical college, 51.2% [10] of the participants were satisfied. However, the magnitude of students satisfied with CLE in this study was lower than that in studies conducted in Namibia,65.9% [22], in Rwanda, 58% [23] and Saud Arabia, 81.81% [24]. These discrepancies may be due to differences in the level of student's study years, study settings, sample size and method of data analysis. On the other hand, the finding of this study is higher when compared to the study done in Ethiopia, Wolkite university, in which 39.9% [5]. This inconsistence may be due to differences in sampling techniques and study settings; the previous research was done only at one university. Also, the finding of the present study is higher than the study conducted in Dares Salaam at University of Dodoma in which 20.1% of nursing students satisfied to CLE [25]. This difference may be due to differences in the sample size. The previous study done on 383 participants. In the present study, among sub-dimensions of the CLE nursing students highly satisfied with Supervisory relationship (49.8%) and the Leadership of the ward manager was the lowest satisfied by nursing students (43.7%). which is supported by the findings of the study done in Nepal [10]. Contrary to the present study, a study done in Hindi at New Delhi showed that within the mean score, the leadership style (1.44) is the highest and supervisory relationship (1.29) was the lowest satisfactory among sub dimensions of CLE for nursing students [26].This difference may be due to sampling technique and sample size; the previous research was done only on 90 nursing students selected by convenient sampling method. Also, the finding of the study done in Malaysia at Kebangsaan Malaysia Medical Centre demonstrate that the most satisfactory area for student nurses was the leadership style of the ward manager (72.0%) among sub dimensions of CLE [11]. This might be due to studies considering nursing students in different year levels. 36 In the present study, nursing students who met their clinical objectives of the placement were about 2.742 times more likely to be satisfied with CLE as compared to those who not met. This is supported by a study

done in Tanzania, Dares Salaam at the University of Dodoma, in which those who agreed that they achieved the clinical objectives of the placement were six times more likely to be satisfied with CLE than those who did not agree [20]. But present study contrast with a study done at the Malaysia University of Kebangsaan Medical Centre, in which there was no relationship between the achievements of learning outcomes in the last clinical placement [11]. This inconsistency may be due to differences in the method of data analysis (using one- way analysis of variance by previous study), inadequate supportive staff and material, and students oriented to the wards that were not related to their clinical objective. Also, participants who met three times per week with their clinical preceptors during the latest clinical placement were 2.831 times more likely to be satisfied than those who met 1–2 times per week. This is similar to a study done in Cyprus [2].But it contrasts with the study done in Ethiopia at Wolkite University [5] and Nepal, Medical Colleges of Chitwan [10]. The reason behind this difference may be due to differences in the study area, and the clinical preceptor is not frequently available in similar way in clinical setting of the study area In addition, the supporting staff was associated with the satisfaction level of nursing students to CLE. Those participants who have inadequate supporting staff in the clinical setting were about 86.4% less likely to be satisfied than those whose clinical setting have adequate supporting staff. In contrast, the study done in Zambia showed that in a learning environment whose learners have inadequate clinical supporting staff were about 70% less likely to be satisfied than those whose learning environment has adequate support staff [6]. The possible justification of this difference is differences in sample size and study population; previous study using larger sample size than current study and include only final year nursing students).

## Conclusion

Effective nursing education programs must be created to improve clinical practice satisfaction and support good emotional regulation in nursing students. The findings of this study have shown that below half of the nursing students were satisfied with their Clinical Learning Environment. However, the magnitude of nursing student's satisfaction in current study was found to be high as compared to recently researched study in Ethiopia. Students who had achieved their clinical objective of the placement and students who received more frequent supervision were more satisfied to the clinical learning environment.

### Limitation of the study

This study only looks at the quantitative data; therefore, additional insights into the qualitative responses from the students regarding their opinions and/or perceptions of the CLE could be gained by using a mixed methodology akin to other pertinent studies. The impact of the CLE on students' learning and development cannot be solely determined by looking at student satisfaction. Other intriguing and unknown variables may surface in such a mixed methodology. The study was limited to only public universities, the findings from a private facility may be different. Furthermore, the questionnaire contained Likert scale questions that offered a choice to participants to be indecisive when responding. This aspect might have affected the accuracy and validity of the study findings

### Recommendations

The following recommendations are forward to the concerned bodies based on the above result;-

To the Nursing Department: Nursing department orient adequate supervisor and work with hospital head nurse by develop strategies on improving students' satisfaction to CLE. In addition, follow up availability of oriented clinical preceptor in the clinical setting at working time by communicate with students.

To the clinical preceptor: In order to achieve student's clinical objective of the placement the clinical preceptor should be orient the students to their clinical objective at a time. Also, clinical teachers avail themselves within clinical setting frequently.

**To minister of education:** Minister of education should assign adequate clinical preceptor for university and provide necessary materials for clinical setting where students clinical attachment holds by cooperate with minister of health.

## Acknowledgments

Our special gratitude extends to Ambo university colleague of medicine and health sciences for giving us this opportunity. We would also like to extend my appreciation to Health Science College particularly department of nursing for their giving information about students' information and the study participants.

## Author contributions

**Project administration:** Adamu Birhanu, Jiregna Darega.

**Software:** Meseret Robi Tura.

**Writing – original draft:** Birhanu Tesfaye.

**Writing – review & editing:** Samuel Abate.

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
