## [Decision Letter · Decision Letter 0]

Dear Dr. Abate,

Thank you for submitting your manuscript to PLOS ONE. After careful consideration, we feel that it has merit but does not fully meet PLOS ONE’s publication criteria as it currently stands. Therefore, we invite you to submit a revised version of the manuscript that addresses the points raised during the review process.

Please submit your revised manuscript by  Jul 12 2024 11:59PM. If you will need more time than this to complete your revisions, please reply to this message or contact the journal office at plosone@plos.org . A rebuttal letter that responds to each point raised by the academic editor and reviewer(s). You should upload this letter as a separate file labeled 'Response to Reviewers'.A marked-up copy of your manuscript that highlights changes made to the original version. You should upload this as a separate file labeled 'Revised Manuscript with Track Changes'.An unmarked version of your revised paper without tracked changes. You should upload this as a separate file labeled 'Manuscript'.

We look forward to receiving your revised manuscript.

Kind regards,

Maria José Nogueira, Ph.D.

Academic Editor

PLOS ONE

Journal Requirements:

Additional Editor Comments:

This manuscript is an important contributor to nursing teaching

Recommendations:

The manuscript has 4 tables, and for all of them I suggest:

1- synthesize the information to make it more appealing and for clearer reading;

2- reduce information from the table titles (if absolutely necessary, place it as a table note -footnote);

3- Do not use redundant information in the table titles or the content of the table vs text

Reviewers' comments:

Reviewer's Responses to Questions

**Comments to the Author**

1. Is the manuscript technically sound, and do the data support the conclusions?

Reviewer #1: Yes

2. Has the statistical analysis been performed appropriately and rigorously?

Reviewer #1: Yes

3. Have the authors made all data underlying the findings in their manuscript fully available?

Reviewer #1: Yes

4. Is the manuscript presented in an intelligible fashion and written in standard English?

Reviewer #1: Yes

Reviewer #1: Level of Satisfaction to Clinical Learning Environment and Its Associated Factors among Nursing Students of Public Universities in Central Ethiopia

Congratulations on choosing this topic; it is pertinent and relevant for understanding the clinical learning of students and the conditions that influence their satisfaction. Utilizing a cross-sectional study is highly appropriate for deepening knowledge in this area.

Abstract

Regarding the abstract, the objective is concise and appropriate to the topic and nature of the study. It summarizes the most relevant aspects without exceeding 300 words. It clarifies who the study participants are and the context.

Introduction

The introduction is well-supported by reference authors. The references are extensive, current, and the topic is explored adequately.

Materials and Methods

1. Clarifies the type of study;

2. Identifies the type of context;

3. Clearly explains the criteria for sample identification and inclusion;

4. Ethical considerations for a study of this nature are clear.

Limitations and Constraints

It is essential to present the study's limitations and constraints, as well as the benefits of these findings for the education and preparation of nurses.

Conclusions

In a study of this nature, the conclusions could be richer and more robust, clearly highlighting the most relevant findings of the study.

Tables and Graphs

Regarding the graphs and tables, they should have a more appealing graphical presentation to improve the understanding of the results.

**Do you want your identity to be public for this peer review?** For information about this choice, including consent withdrawal, please see our Privacy Policy

Reviewer #1: No

---

## [Author Response · Author response to Decision Letter 1]

8 Aug 2024

I have read it and I have tried to use as much as possible

We suggest you thoroughly copyedit your manuscript for language usage, spelling, and grammar. If you do not know anyone who can help you do this, you may wish to consider employing a professional scientific editing service.

We note that you have indicated that there are restrictions to data sharing for this study. For studies involving human research participant data or other sensitive data, we encourage authors to share de-identified or anonymized data. However, when data cannot be publicly shared for ethical reasons, we allow authors to make their data sets available upon request.

b) If there are no restrictions, please upload the minimal anonymized data set necessary to replicate your study findings to a stable, public repository and provide us with the relevant URLs, DOIs, or accession numbers. Please see for guidelines on how to de-identify and prepare clinical data for publication. For a list of recommended repositories, please see. You also have the option of uploading the data as Supporting Information files, but we would recommend depositing data directly to a data repository if possible.

Response:-all the data used for this research were included in the document. And the the raw data can be accessed from corresponding author at a reasonable request.

The manuscript has 4 tables, and for all of them I suggest:

1- synthesize the information to make it more appealing and for clearer reading;

2- reduce information from the table titles (if absolutely necessary, place it as a table note -footnote);

3- Do not use redundant information in the table titles or the content of the table vs. text

Response: - Revised accordingly

Reviewers' comments:

Reviewer's Responses to Questions

Comments to the Author

1. Is the manuscript technically sound, and do the data support the conclusions?

Reviewer #1: Yes

2. Has the statistical analysis been performed appropriately and rigorously?

Reviewer #1: Yes

3. Have the authors made all data underlying the findings in their manuscript fully available?

Reviewer #1: Yes

4. Is the manuscript presented in an intelligible fashion and written in standard English?

Reviewer #1: Yes

5. Review Comments to the Author

Reviewer #1: Level of Satisfaction to Clinical Learning Environment and Its Associated Factors among Nursing Students of Public Universities in Central Ethiopia

Congratulations on choosing this topic; it is pertinent and relevant for understanding the clinical learning of students and the conditions that influence their satisfaction. Utilizing a cross-sectional study is highly appropriate for deepening knowledge in this area.

Abstract

Regarding the abstract, the objective is concise and appropriate to the topic and nature of the study. It summarizes the most relevant aspects without exceeding 300 words. It clarifies who the study participants are and the context.

Introduction

The introduction is well-supported by reference authors. The references are extensive, current, and the topic is explored adequately.

Materials and Methods

1. Clarifies the type of study;

2. Identifies the type of context;

3. Clearly explains the criteria for sample identification and inclusion;

4. Ethical considerations for a study of this nature are clear.

Response:-. Type of the study: - An institutional based cross-sectional study involving was conducted

Response: - Clearly explains the criteria for sample identification and inclusion;-

Response: - Study population

All randomly selected second year and third year undergraduate nursing students who are studying at universities found in Central Ethiopia.

Response:-Sampling unit

Individual Nursing student who was selected during the data collection period.

Response:-Inclusion criteria

All undergraduate nursing students of second year and third year as well as those participated at least once in clinical practice.

Response:-Exclusion criteria

Fourth year nursing student due to educational leave at the time of the study.

Limitations and Constraints

It is essential to present the study's limitations and constraints, as well as the benefits of these findings for the education and preparation of nurses.

Response: - This study only looks at the quantitative data; therefore, additional insights into the qualitative responses from the students regarding their opinions and/or perceptions of the CLE could be gained by using a mixed methodology akin to other pertinent studies. The impact of the CLE on students' learning and development cannot be solely determined by looking at student satisfaction. Other intriguing and unknown variables may surface in such a mixed methodology. The study was limited to only public universities, the findings from a private facility may be different. Furthermore, the questionnaire contained Likert scale questions that offered a choice to participants to be indecisive when responding. This aspect might have affected the accuracy and validity of the study findings.

Response: - Benefits of these findings for the education and preparation of nurses.

The finding of this study will be help nursing department, minister of education and nurse association to understand the nursing student’s satisfaction with clinical learning environment and factors that affect nurse students’ satisfaction and to design strategy to enhance factors positively influence on nurses ‘student satisfaction as well as improve factors that contribute for dissatisfaction with clinical learning environment. In addition, the knowledge generated from this study will be enriched literatures available on the issue and aid for further a researcher who has an interest in conducting researches on this topic. ________________________________________

6. PLOS authors have the option to publish the peer review history of their article (what does this mean?). If published, this will include your full peer review and any attached files.

Do you want your identity to be public for this peer review? For information about this choice, including consent withdrawal, please see our Privacy Policy.

Reviewer #1: No

# I have done it accordingly

---

## [Decision Letter · Decision Letter 1]

Dear Dr. Abate,

We look forward to receiving your revised manuscript.

Kind regards,

Maria José Nogueira, Ph.D.

Academic Editor

PLOS ONE

Dear authors

Please respond to all suggestions made by reviewers, in order to overcome the weaknesses highlighted in the manuscript, especially those highlighted by reviewer 2.

Reviewers' comments:

Reviewer's Responses to Questions

**Comments to the Author**

Reviewer #2: All comments have been addressed

2. Is the manuscript technically sound, and do the data support the conclusions?

Reviewer #2: Yes

3. Has the statistical analysis been performed appropriately and rigorously?

Reviewer #2: Yes

4. Have the authors made all data underlying the findings in their manuscript fully available?

Reviewer #2: Yes

5. Is the manuscript presented in an intelligible fashion and written in standard English?

Reviewer #2: Yes

Reviewer #2: The starting question of the study is not clear when reading the document, nor is the objective. I suggest that these items, which are essential to all studies, be revised.

The presentation of the results is descriptive, which makes it confusing. My suggestion is that it should be presented using tables and graphs that are appealing for reading and interpreting the data, thus making it easier to understand.

The table titles are too big, they need to be improved.

**Do you want your identity to be public for this peer review?** For information about this choice, including consent withdrawal, please see our Privacy Policy

Reviewer #2: **Yes: ** Delfina Ana Pereira Ramos Teixeira

---

## [Author Response · Author response to Decision Letter 2]

6 Nov 2024

Editor Comments:

Dear authors

Please respond to all suggestions made by reviewers, in order to overcome the weaknesses highlighted in the manuscript, especially those highlighted by reviewer 2.

Response: We appreciate the valuable suggestions and recommendations given to us by the editor and the reviewers. In this revision, due attention was given to every section, especially the comments given by reviewer 2. All comments point by point were addressed and highlighted by yellow color in the revised content in the text. Thank you!

Reviewer #2:

1. The starting question of the study is not clear when reading the document, nor is the objective. I suggest that these items, which are essential to all studies, be revised.

Response: Thank you for your suggestions. This has now been corrected as suggested. Under the introduction section, page 5, paragraph 3, lines 126-139.

2. The presentation of the results is descriptive, which makes it confusing. My suggestion is that it should be presented using tables and graphs that are appealing for reading and interpreting the data, thus making it easier to understand.

Response: Thank you for your suggestion and comment. This has now been corrected as suggested. Under references section page 13, line 343 and page 11 line 303, page 12, line 321, page 12 line 325.

3. The table titles are too big, they need to be improved

Response: We sincerely acknowledged and improved it. This has now been corrected as suggested. Under result section, Page 21-26, line 527-539.

---

## [Editor Report · Decision Letter 2]

Level of Satisfaction to Clinical Learning Environment and Its Associated Factors among Nursing Students of Public Universities in Central Ethiopia

PONE-D-24-08053R2

Dear Dr. Samuel Abate

We’re pleased to inform you that your manuscript has been judged scientifically suitable for publication and will be formally accepted for publication once it meets all outstanding technical requirements.

Kind regards,

Maria José Nogueira, Ph.D.

Academic Editor

PLOS ONE

Additional Editor Comments (optional):

The authors made the suggested corrections to the manuscript, which greatly improved it.

The manuscript is now ready to be accepted for publication.
---

## [Editor Report · Acceptance letter]

PONE-D-24-08053R2

PLOS ONE

Dear Dr. Abate,

I'm pleased to inform you that your manuscript has been deemed suitable for publication in PLOS ONE. Congratulations! Your manuscript is now being handed over to our production team.

Kind regards,

on behalf of

Professor Maria José Nogueira

Academic Editor

PLOS ONE